# OpenReview forum: "SWE-Lancer: Can Frontier LLMs Earn $1 Million from Real-World Freelance Software Engineering?"
_ICML.cc/2025/Conference — ICML 2025 oral_

### Official Review · Reviewer_h9KZ · 2025-03-06

**Overall Recommendation:** 4

**Summary:**

The paper introduces a benchmark of 1488 software engineering freelance tasks consisting of managerial and coding tasks to evaluate LLM performance against real world tasks. The benchmark helps get a clearer picture about the potential social and economic impacts of AI. For individual contributor/coding tasks, comprehensive e2e tests are developed and a user tool is provided to mimic developer's actions associated with the task in a browser.
Performance of various LLMs is measured as Pass@1, earning, and earning rate. Reported results show a significant gap between SOTA LLMs' performance and what human can achieve. This gap is more spelled out for individual contributor tasks where pass@1 remains below 30% versus managerial tasks for which this value gets up to 45%.

**Claims And Evidence:**

The main claim of the paper is raising more awareness about LLMs potential in automating real-world software engineering tasks to get a clearer picture of their social and economic impacts. The claim is clear and is backed up by the reported experimental results.

**Essential References Not Discussed:**

There are certain relevant and seminal works that are not cited in the paper. For example, Evaluating Large Language Models Trained on Code by Chen et al. and Program Synthesis with Large Language Models by Austin et al. are among such cases.

**Experimental Designs Or Analyses:**

The experimental design looks sound, e.g. authors are mindful of the possible contamination issues and have some mitigation plans. The analysis accompanied the experimental results including the ablation studies looks sound as well.

**Methods And Evaluation Criteria:**

The prepared benchmark based on Upwork and the evaluation criteria to gauge the LLM performance makes sense.

**Other Comments Or Suggestions:**

No other comments/suggestions.

**Other Strengths And Weaknesses:**

**Strengths**

- The comprehensive benchmark of real-world software engineering tasks with curated e2e tests that are tied with monetary rewards is a valuable asset for the research community. On top of that, the experimental results and providing a realistic picture of AI's capability in this domain is extremely valuable and will trigger valuable discussions in the research community.

**Weaknesses**
- The weakest aspect of the paper is the literature review section. Authors could have done a more comprehensive study of the existing literature. More importantly, they could have compared their learning from reported experiments in their work against some of the claims or experimental results reported in other existing works.

**Questions For Authors:**

- After reading the paper I'm still not clear how the user tool is helping. The paper says "The model does not receive any feedback about whether the action was successful" it also says "a text-based trajectory and a series of screenshots are written to its working directory when the tool finishes.". Should I assume by "action was successful" authors mean whether e2e test passes successfully?

- Still related to user tool, based on Figure 9, it looks like user tool is not helping GPT-4o much, can authors explain why that's the case for GPT-40 but not O1.

- One thing that I could not follow is whether managerial tasks are also collected from Upwork, the idea of having a few proposals and letting LLM decide which one is more appropriate is not aligned with my understanding of the tasks on a platform like Upwork. Maybe my knowledge is limited but can authors clarify?

**Relation To Broader Scientific Literature:**

Studying AI/LLM performance against real-world tasks that are tied with monetary reward is a critical and relevant topic to the broader scientific literature since they can adjust expectations about what can versus cannot be expected from AI and ground everyone's perception of AI's potential.

**Theoretical Claims:**

There is not much theoretical claim in the paper so this box is less applicable.

---

> ### Author Rebuttal · Authors · 2025-04-01
>
> Thank you very much for your careful and thoughtful review! We’ll address your points below.
>
> > There are certain relevant and seminal works that are not cited in the paper. For example, Evaluating Large Language Models Trained on Code by Chen et al. and Program Synthesis with Large Language Models by Austin et al. are among such cases.
>
> Thank you for this flag. We do include Chen et al.’s paper, HumanEval, in the Related Work section, but we will cite directly and add to Table A.4. We sincerely apologize for not citing Austin et al. and will include it in the camera-ready version if accepted to the conference (as per ICML rules we cannot update the submitted PDF in response to rebuttals).
> To be more comprehensive, we also plan to add:
> - Lu et al. (2021) – CodeXGLUE
> - Li et al. (2022) –  AlphaCode
> - Nijkamp et al. (2022) – CODEGEN
> - Lai et al. (2023) – DS-1000
>
> Please let us know if there are others we should include. We sincerely apologize for our oversight and appreciate you pointing this out.
>
> > More importantly, they could have compared their learning from reported experiments in their work against some of the claims or experimental results reported in other existing works.
>
> Great point! In terms of comparing learnings:
> - For IC SWE tasks, one comparable benchmark is SWE-Bench Multimodal, where GPT-4o solved 12.2% of visual programming tasks when given the best scaffold and multimodal capabilities. This is comparable to our GPT-4o scoring 8.0% on our IC SWE Diamond tasks when the model can ingest image screenshots via the user tool.
> - On SWE-Bench Verified, the industry standard benchmark for software engineering, GPT-4o currently scores 38.8% using the best available scaffold, but only 8% on SWE-Lancer Diamond IC SWE – likely because SWE-Bench Verified focuses on bugfixes, and SWE-Lancer includes open-ended problems (adding new features, reliability) and full-stack development. This highlights the need for more challenging, unsaturated SWE benchmarks.
> - SWE-Lancer is unique in evaluating engineering management skills – we did not identify a direct analogue in prior works; we welcome any suggestions for related work.
> We will include these comparisons in the camera-ready version of our paper if accepted to ICML.
>
> > After reading the paper I'm still not clear how the user tool is helping. The paper says "The model does not receive any feedback about whether the action was successful" it also says "a text-based trajectory and a series of screenshots are written to its working directory when the tool finishes.". Should I assume by "action was successful" authors mean whether e2e test passes successfully?
>
> We’re happy to clarify! When the model calls the user tool, it automatically performs actions relevant to the task on the website (specifically, by using the Playwright library to simulate user actions), and sends the model a report (screenshots + a text-based log of browser states). The user tool does NOT provide feedback to the model on whether it passed or failed the end-to-end tests, but it does allow for iterative debugging and verification of hypotheses. We created the user tool to mirror how humans iteratively check their code by running it and inspecting the output.
>
> > Based on Figure 9, it looks like user tool is not helping GPT-4o much, can authors explain why that's the case for GPT-40 but not O1.
>
> Definitely! Smarter models are able to “get more out of”  the user tool. GPT-4o often fails to properly use the user tool. For instance, as we outline in section A.9. of the appendix, the model fails to review the user tool logs that indicate the actual behavior.
>
> > One thing that I could not follow is whether managerial tasks are also collected from Upwork, the idea of having a few proposals and letting LLM decide which one is more appropriate is not aligned with my understanding of the tasks on a platform like Upwork.
>
> Happy to clarify! The tasks in this benchmark were posted on both Upwork and GitHub. When a task is posted (e.g., “solve X bug”), interested freelance software engineers are required to first submit proposals for solving the task on the corresponding Github issue (see the Expensify/App GitHub repository for examples). A freelance engineering manager then reviews all proposals and chooses the best one. The engineer whose proposal is chosen then gets to actually implement the solution. For each task with multiple proposals, we created a “managerial task” by collecting all the freelancer proposals, having models choose the best proposal, and comparing model choices to the real manager’s. You are right that not all Upwork software engineering tasks entail a proposal selection process – this just reflects Expensify’s public vetting process.
>
>  Thank you again for your valuable insights. We hope we have addressed your concerns and respectfully invite you to consider raising your score if you feel the benchmark has improved as a result of these clarifications.

---

> > ### Comment · Reviewer_h9KZ · 2025-04-02
> >
> > I'd like to thank the authors for addressing my comments, providing further clarifications/explanations, and their planned updates for the related work section. I keep my "accept" recommendation.

---

### Official Review · Reviewer_ue7i · 2025-03-12

**Overall Recommendation:** 4

**Summary:**

This paper introduces SWE-Lancer, a benchmark for evaluating language models' capabilities in real-world software engineering tasks. The benchmark comprises 1,488 freelance software engineering tasks from Upwork, collectively valued at $1 million USD in actual payouts. SWE-Lancer includes two distinct task categories: (1) Individual Contributor (IC) SWE tasks where models generate code patches to resolve real-world issues, and (2) SWE Manager tasks where models select the optimal implementation proposal from multiple options. Unlike previous benchmarks, SWE-Lancer evaluates performance across the full technology stack with end-to-end tests created by professional software engineers.

**Claims And Evidence:**

The claims made in the submission are well-supported by evidence:

- The claim that SWE-Lancer provides a more realistic evaluation of SE capabilities is supported by the detailed description of task selection, construction, and verification processes. The use of real freelance tasks with actual monetary values and end-to-end testing provides stronger evidence than previous benchmarks.

- The performance claims for different models are backed by experimental results presented in tables and figures, showing pass rates and earnings across different task types, categories, and conditions.

**Essential References Not Discussed:**

The paper appears to cover the relevant prior work in software engineering benchmarks and LLM evaluation.

**Experimental Designs Or Analyses:**

- The comparison across multiple frontier models (GPT-4o, o1, Claude 3.5 Sonnet) provides a good overview of current capabilities.

- The ablation studies on reasoning effort, number of attempts, and user tool access offer insights into factors affecting performance.

- The reporting of both pass rates and earnings gives a more complete picture of model performance.

**Methods And Evaluation Criteria:**

The proposed methods and evaluation criteria are appropriate for the problem at hand:

- The division into IC SWE and SWE Manager tasks allows for evaluation of different aspects of software engineering capabilities.

- Using real-world freelance tasks with actual monetary values provides a economic significance to the evaluation.

- The end-to-end testing approach is a significant improvement over unit tests used in previous benchmarks.

- The evaluation of models under different conditions (reasoning effort, number of attempts, with/without user tool) provides insights into factors affecting performance.

**Other Comments Or Suggestions:**

The paper would benefit from a more detailed discussion of the agent framework in the main text rather than relegating it to the appendix.

**Other Strengths And Weaknesses:**

Strengths:

**Real-world relevance**: Using actual freelance tasks with real monetary values provides a realistic evaluation of LLM capabilities.

**End-to-end testing**: The use of comprehensive end-to-end tests rather than unit tests is a improvement over previous benchmarks.

**Economic dimension**: Mapping performance to monetary value adds an important perspective on the practical impact of AI capabilities.

**Comprehensive evaluation**: The paper evaluates multiple models under various conditions, providing a rich understanding of current capabilities and limitations.

Weaknesses:

**Limited description of the agent framework**: Although the details of the agent framework are mentioned in the appendix of the paper, it is still difficult to fully understand its implementation.

**Questions For Authors:**

- Have you analyzed whether there are systematic patterns in the types of tasks that models fail to solve correctly? This could provide valuable insights for targeted improvements in model capabilities.

- Given that SWE Manager tasks showed higher success rates than IC SWE tasks, do you believe this reflects a genuine difference in capability, or could it be related to the evaluation methodology (selection from options vs. generation)?

- How might the benchmark be extended to evaluate human-AI collaboration more directly, beyond the current user tool experiments? This seems like an important direction given that collaborative scenarios may be more realistic in practice than fully autonomous AI software engineers.

**Relation To Broader Scientific Literature:**

It connects to broader research on automated software engineering, agentic systems, and the economic impact of AI.

**Theoretical Claims:**

The paper does not make theoretical claims that require formal proofs.

---

> ### Author Rebuttal · Authors · 2025-04-01
>
> Thank you very much for your careful and thoughtful review! We’ll address your points below.
>
> > Limited description of the agent framework: Although the details of the agent framework are mentioned in the appendix of the paper, it is still difficult to fully understand its implementation. The paper would benefit from a more detailed discussion of the agent framework in the main text rather than relegating it to the appendix.
>
> Thank you for flagging! We included information about our agent harness in the appendix to satisfy the ICML page limit requirements. We’ve also open sourced eval-harness code so anyone can plug in an API key and run our eval. We are happy to update the main paper with more details about the agent framework in the methodology section in the camera-ready version if helpful. Are there specific aspects of the agent framework you’d like us to share details on? We are happy to clarify them here if helpful!
>
> > Given that SWE Manager tasks showed higher success rates than IC SWE tasks, do you believe this reflects a genuine difference in capability, or could it be related to the evaluation methodology (selection from options vs. generation)?
>
> Great question! Our manager tasks involve selecting from predefined options, and the base rate of selecting a correct answer by chance in a multiple-choice setting is higher than that for an open-ended question. Therefore, the “lower bound” for what a model could achieve is by definition higher in our management tasks than our IC tasks that involve writing code. Still, the management success rate on our evaluation is much higher than can be attributed to chance alone, and the difference in success rates is meaningful (management success rates are more than double that of IC SWE tasks), suggesting there is also a capability difference at play. Qualitative analysis of trajectories also indicates that models we tested are stronger in conceptual thinking required for management tasks versus in full hands-on coding, where they are more likely to make a mistake. A potential future improvement to our evaluation is to create a more open-ended management test to ensure measured differences are due to capability; however, given that the actual tasks for managers in real life in our dataset required managers to select proposals from predefined options, we kept this grading schema to better match reality. It’s also worth noting that manager tasks in SWE-Lancer do not capture all management skills, which also include defining project timelines, deciding resource allocation, and making strategic/organizational plans.
>
> > How might the benchmark be extended to evaluate human-AI collaboration more directly, beyond the current user tool experiments? This seems like an important direction given that collaborative scenarios may be more realistic in practice than fully autonomous AI software engineers.
>
> Excellent question - we agree this is an important direction for future work! We have open-sourced our evaluation harness and the diamond split of our dataset, and we encourage the community to build upon this to further explore human-in-the-loop approaches. We regret that we had limited time and resources while building the evaluation, and were not able to explore this exciting direction.
>
> Thank you again for your valuable insights. We hope we have addressed your concerns and respectfully invite you to consider raising your score if you feel the benchmark has improved as a result of these clarifications.

---

> > ### Comment · Reviewer_ue7i · 2025-04-02
> >
> > Thank you for the author's response. I will maintain my original positive score!

---

### Official Review · Reviewer_E3n6 · 2025-03-17

**Overall Recommendation:** 5

**Summary:**

This paper introduces SWE-Lancer, a benchmark of 1488 real-world freelance software engineering tasks from Upwork valued at $1million USD in actual payouts. The benchmark includes both Individual Contributor (IC) tasks where models generate code patches to fix issues, and Software Engineering Manager tasks where models select the best implementation proposal. Tasks are evaluated using end-to-end tests with experiments showcasing that even the best performer (Claude 3.5 Sonnet) solves only 26.2% of IC tasks and 44.9% of Manager tasks on the public Diamond set, indicating these real-world tasks remain challenging despite recent progress in coding capabilities.

**Claims And Evidence:**

The key claims are well-supported by the comprehensive methodology and results! However, I would appreciate if the authors could ammend:

1. In Line 134 to 144 the authors claim, that previous benchmarks were biased (eg, SWE-Bench involved selecting PRs with unit tests), while SWE-Lancer follows "unbiased data collection." However, this doesn't accounts for already biased distribution of tasks present in Expensify and UpWork in general (line 406-411). Therefore, the same should be clarified upfront while describing the SWE-Lancer unbiased.

**Essential References Not Discussed:**

NA

**Experimental Designs Or Analyses:**

The experimental design is rigorous with comprehsnive evaluation on multiple frontier models. A minor limitation however is that the benchmark draws exclusively from one repository (Expensify) and one platform (Upwork), which may limit generalizability across different domains of software engineering.

**Methods And Evaluation Criteria:**

The evaluation approach is sound:

1. The use of end-to-end tests rather than unit tests is particularly valuable, as it replicates real-world behavior of human reviewers.
2. Mapping model performance to real monetary value provides a meaningful economic context that goes is more meaningful than accuracy metrics.
3. Including management tasks addresses an important aspect of software engineering that previous benchmarks have neglected.

**Other Comments Or Suggestions:**

The paper is well-written and is a valuable contribution to improve SWE agents further. However, it would have been great, if authors had made specific efforts in increasing the accessibility of the dataset, given the high cost required to evaluate on SWE-Lancer.

**Other Strengths And Weaknesses:**

Strengths:

1. The benchmark's economic valuation framework provides a novel and intuitive way to assess the practical impact of model capabilities.
2. The inclusion of both implementation and management tasks offers a more generalizing view and evaluation of software engineering abilities.
3. The rigor in experiments and analysis is commendable.
4. The authors open-source the benchmark and environment, which is great for future research.

Weaknesses:

Apart from points mentioned above,
1. The end-to-end tests may still have limitations, such as them being static, with limited guarantees on coverage.
2. Similar to prior benchmarks, the benchmark still primarily contains questions related to bug fixing (~90% of tasks), compared to new features, maintenance, etc. (Table 3)
3. The benchmark is very expensive to run in terms of computational cost and time, limiting immediate adaptation by the community. Any specific efforts to address the issue would have made the impact of SWE-Lancer stronger.

**Questions For Authors:**

1. How might the distribution of tasks in SWE-Lancer differ from those encountered by full-time software engineers rather than freelancers? Does this affect the benchmark's representativeness of general software engineering work?
2. What analysis have the authors done on the characteristics of tasks that models find particularly challenging versus those they solve more easily? Are there patterns in task types, or importantly domains that predict model performance?
3. In prompts, authors provide "The reward for the correct fix is ${price}." Given prior research has shown artificially incentivizing LLMs with monetary rewards can improve their performance, have authors evaluated artifically varying price to see change in model performance? For instance, could performance on easier tasks, been decreased simply because of lower amounts provided in the prompt?
4. How does the benchmark account for multiple valid solutions to the same problem? Especially for more complex tasks, there may be various approaches with different trade-offs that could all satisfy the end-to-end tests?
5. Have authors seen cases, where the models pass the end-to-end tests, but are still not fully correct?
6. Have you considered extending the benchmark to include aspects of software engineering beyond correctness, such as code quality, maintainability, or adherence to best practices? If not, this should be part of limitation, so the community does not make incomplete conclusions from the paper.
7. In Lines 620-621, authors mention: "If these three categorizations were not unanimous, we manually assigned the final categories." it would be interesting to see how often the manual verification was done?

**Relation To Broader Scientific Literature:**

The paper thoroughly positions SWE-Lancer relative to existing benchmarks articulating clearly how SWE-Lancer differs from previous benchmarks like SWE-Bench by focusing on real-world, full-stack commercial engineering rather than isolated programming tasks. The comprehensive comparison table in Appendix A4 also clearly highlights the unique contributions of SWE-Lancer.

**Theoretical Claims:**

NA

---

> ### Author Rebuttal · Authors · 2025-04-01
>
> Thank you for your generous review and strong endorsement! Below we address your points.
>
> > Unbiased data collection vs. platform bias
>
> Great point! SWE-Lancer  focuses on a single repository and freelance tasks, so it isn’t free from platform bias. By ‘unbiased data collection,’ we mean we didn’t selectively pick tasks or rely on pre-written tests; we took tasks as posted on Upwork and had professional engineers write E2E tests. We will clarify that ‘unbiased data collection’ refers to minimal artificial filtering, not the absence of any underlying biases.
>
> > Most tasks are bugfixes
>
> This reflects the distribution of the real-life Expensify repo, as it is a mature production system where most tasks involve debugging and incremental improvements. With SWE-Lancer, we set out to ‘containerize’ a company in an unopinionated manner, so we didn’t filter for task type or ease of testing. We acknowledge that one drawback is that the distribution may not be ideal or interesting; however, it ensures that our dataset is realistic.
>
> > Benchmark is expensive to run
>
> We recognize that running SWE-Lancer is resource-intensive, as there are 1488 tasks which each run in their own VM. We open-sourced our evaluation harness so researchers can select smaller subsets to manage cost (e.g., some researchers run 5–10 SWE-Lancer tasks locally). In response to your feedback, we have also updated our open-sourced repository with a SWELancer-Lite split with 174 tasks (vs. the 502 originally in the Diamond set).
>
> > SWE-Lancer task distribution
>
> Good question! We note in the limitations section that freelancer work tends to be more self-contained and well-scoped than full-time software engineering work. Furthermore, SWE-Lancer would be less representative of SWE work that is going “zero to one,” as even the tasks involving adding a new feature are building off of a well-established and maintained codebase. We will be sure to elaborate on this in the camera-ready version.
>
> > Characteristics of challenging tasks
>
> We analyzed pass rates by domain and task types (Tables 2 and 3 of paper) as well as individual rollouts. Overall, models performed much better on SWE Manager tasks than IC SWE tasks, likely because qualitative analysis is easier than execution. Models tend to perform better on IC SWE bugfix tasks as those are easier to root-cause vs. more open-ended tasks (adding new features) or reliability changes that span more of the codebase. All models were worse at IC SWE tasks involving application logic, and GPT-4o and o1 in particular performed worse than Claude 3.5 Sonnet on UI/UX tasks.
>
> > Artificial variation of price in prompt
>
> Great question. We found that omitting price information did not materially impact performance, so we decided not to pursue this direction further; however, in response to your feedback, we will directly evaluate sensitivity to the price offered in prompt as a new experiment for the camera-ready paper, if accepted.
>
> > Multiple valid solutions to the same problem
>
> This is a great question, and a challenge for SWE benchmarks. To address this, we employ end-to-end tests, which are more flexible than unit tests because they are agnostic to the specific implementation methodology and pass as long as the end behavior is correct. Of course, the model’s implementation could still deviate from the tests in a way that they fail to account for. To mitigate this, we allow the model to invoke a User Tool, which opens the browser and performs the action associated with the task (e.g. uploading a profile picture). You can read more about the User Tool in section 2.2.5 of our paper.
>
> > Cases where models pass E2E tests but are still not fully correct
>
> We have manually reviewed rollouts and haven’t observed this; however, the possibility of cheating and/or grading errors is a concern for every benchmark, so we cannot rule this out. We find that end-to-end tests are significantly harder to cheat on compared to unit tests, as it is significantly harder to fake correct behavior when the entire system is involved. We include examples of unit test hacking occurring in SWE-Bench in the Appendix to illustrate some of the advantages of end-to-end testing.
>
> > Extending beyond correctness
>
> We agree this is an exciting future direction, and will mention this explicitly under Future Work for the camera ready version of our paper. We did not want to make the evaluation too complex to run, nor the results too difficult to interpret, and imbuing notions of code quality and maintainability into the benchmark would have contributed significantly to its complexity. We believe that determining the best way to test for these is an open research question, and would encourage others to leverage our dataset and open-source codebase to experiment with more complex evaluation setups.
>
> > Frequency of manual assignment
>
> Manual assignment was performed in 15% of cases.
>
> Thank you again for your encouraging feedback!

---

### Official Review · Reviewer_FybT · 2025-03-20

**Overall Recommendation:** 4

**Summary:**

The paper introduces a benchmark SWE-Lancer using 1488 freelancing tasks from Upwork.
The benchmark offers many advantages as compared to the existing SWE benchmark: connects solving an SWE task directly to economic benefits, more challenging problems, a diverse data set including UI/UX tasks.
There are two types of tasks:
i) Individual Contributor (IC) tasks, where models generate code patches to resolve issues
ii) Software Engineering (SWE) Manager tasks, where models select the best implementation proposal
Apart from proposing the benchmark, they tests vLLMs on these tasks and found that the best model, Claude 3.5 Sonnet also resolves only  40.9% of the issues.

**Claims And Evidence:**

Yes, the claims are generally supported by the evidence.
It would be good if authors could do multiple runs and provide confidence intervals. The LLMs are quite stochastic and concluding from a single number is misleading.

**Essential References Not Discussed:**

None

**Experimental Designs Or Analyses:**

Yes, I checked them namely, increasing the number of attempts, test time computation and removing tool usage.

**Methods And Evaluation Criteria:**

The evaluation criteria used are pass @1, pass @k and real-world economic benefit.
Yeah, these make sense for the concerned applications.
However it would be better if authors could also propose metrics that can evaluated based on the partially correct trajectory as well, i.e., the issue is not resolved but somewhat have done something reasonable. This will be beneficial for understanding where do current model stands.

**Other Comments Or Suggestions:**

line 86: explain diamond set when it occurs for the first time.

the paper mention about bias in SWEbench due to having unit tests in them; however i believe there is also strong bias is swe-lancer due to only having test based data, that too from expensity. I think getting rid of bias is difficult and my point is to not as a benefit or a disadvantage (line 134:144).

Also, I think on page 1-3, the advantages of swelancer is a bit verbose with some repeats and perhaps can be shortened.

### Update after rebuttal

All my questions have been addressed.
I also agree with the motivation to include managerial tasks.
For authors: Please include the promised changes in the final version.
I have changed the score to accept.

**Other Strengths And Weaknesses:**

Strengths:
- The paper is clearly written and easy to parse
- I believe the contribution of adding more realistic problems is quite strong
- They employed professionals (human annotated data) for writing and verifying end to end tests
- Experiments are conclusive

Weakness:
- tasks are only from the Expensify repository. This limitation it quite strong IMO.
- Would be good to include multimodal data
- Motivation of introducing managerial task is rather weak. I think a benchmark just with IC would also be valuable for community
- The associated costs are dynamically priced, however, in actual deployment LLM's would solve it immediately (for the lowest price) ---- i believe it would not wait for weeks to get price hiked up?

**Questions For Authors:**

Do you know the spit of tasks based on coding language as well? That would be helpful as well.

Do you have hints or trends that you may have observed in execution that shows if existing LLM is trained on similar data?

Did you also try some open-source models on these tasks?

**Relation To Broader Scientific Literature:**

The key contribution of proposing a benchmark of more complex and realistic problems is beneficial for the community.

**Theoretical Claims:**

Experimental paper -- No theoretical claims

---

> ### Author Rebuttal · Authors · 2025-04-01
>
> Thank you for this thoughtful review! We appreciate your feedback and address your points below.
> > Multiple runs and CIs
>
> Excellent point. In response to your comment, we performed 3 runs of GPT-4o and o1 on the IC SWE and SWE Manager Diamond subsets to provide confidence intervals in the camera-ready version if accepted. We are still running Claude (which takes longer due to rate limits). We calculate 95% CIs using the standard bootstrap procedure that resamples model attempts per problem to approximate the metric’s distribution. Our results are below:
> - GPT-4o (IC SWE Diamond): avg pass@1 = 8.0% (95% CI [4.2%, 11.4%])
> - o1 (IC SWE Diamond): avg pass@1 = 17.5% (95% CI [13.1%, 21.9%])
> - GPT-4o (SWE Manager Diamond): avg pass@1 = 35.1% (95% CI [30.7%, 40.0%])
> - o1 (SWE Manager Diamond): avg pass@1 = 40.8% (95% CI [35.3%, 46.0%])
>
> > Partial progress metrics
>
> Great idea! In response to your feedback, we updated the grading and reporting modules of our open source scaffold to report both full and partial credit. We hope this will allow external researchers to experiment with this idea more easily!
>
> > Single repository
>
> We appreciate this concern and recognized it as a key limitation in our paper. Still, we believe our contribution provides a strong foundation for expanding to more repositories in the future. We initially attempted a multi-repository approach but realized we needed to de-risk our process on a single sufficiently diverse repository first. We chose Expensify for its range of apps (e.g., web, iOS, Android, desktop), tasks (UI/UX, application logic, server-side logic, system-wide reliability improvements, managerial tasks), and maturity (Expensify serves millions of customers). Building our benchmark helped us create a scalable task-creation process to tie LLM performance to economically valuable work. By open-sourcing our methodology and code (with modular components that are applicable across repos), we hope community contributors can now expand to additional codebases.
>
> > Multimodal data
>
> Great point! Our environment already supports image screenshots via the User Tool. In response to your comment, we added a utility to the open-source SWE-Lancer codebase to automatically download each task’s video – along with README instructions to enable easier video task creation in the future. Let us know if this addresses your concerns!
>
> > Motivation for managerial tasks
>
> While many believe LLMs will soon automate IC coding tasks, others argue managerial tasks (like oversight of proposed code changes) will still require human supervision (see e.g. https://www.wsj.com/articles/how-ai-tools-are-reshaping-the-coding-workforce-6ad24c86). In the Expensify repository, managers—who review and select the best technical proposals—earn the same as the freelance engineers implementing the solutions, and we wanted to reflect the full range of repository tasks. Results on management tasks are also unique in helping evaluate the “management is protected from LLMs” hypothesis. For those interested exclusively in the IC SWE tasks, we report these metrics separately in our open source code. We will clarify this in the camera-ready version if accepted.
>
> >Dynamic pricing
>
> Great question! You’re right that a sufficiently powerful model might solve certain tasks right away – effectively preventing price hikes altogether. But that’s exactly the disruption our paper sets out to measure. Our benchmark reflects how tasks are priced in today’s freelancer markets, where unresolved jobs see their rates climb over time. By quantifying this status quo and then applying LLMs to it, SWE-Lancer demonstrates the extent of the economic impact these models would have.
>
> > Diamond Set
>
> Thank you. We will clarify earlier that the “Diamond” subset is the publicly released subset worth $500,800 in payouts; the rest is a private holdout.
>
> > Bias
>
> We mitigate test-based bias with human-written E2E tests that mirror realistic user flows, rather than unit tests from the original codebase (biased towards code people chose to write tests for). Still, all benchmarks have biases, so we encourage using multiple real-world benchmarks to capture a broader range of capabilities.
>
> > Language split
>
> All of the coding tasks are in JavaScript, TypeScript, and React or React Native.
>
> > Data contamination
>
> Because these tasks are from a public repository, we checked for data contamination by comparing pass@1 on tasks published before vs. after model knowledge cutoff dates (Table A.2) and found no evidence of memorization.
>
> > Open-source models
>
> We haven’t tested open-source models yet but plan to run Llama and DeepSeek for the camera-ready version. We’ve also open-sourced our evaluation harness and the Diamond Set to allow others to test models.
>
> Thank you again for your valuable insights. We hope we have addressed your concerns and respectfully invite you to consider raising your score if you feel the benchmark has improved as a result of these clarifications.

---

### Decision · Program_Chairs · 2025-05-01

**Decision:**

Accept (oral)

**Comment:**

This paper introduces SWE-Lancer, a novel benchmark for evaluating large language models on software engineering tasks derived from real-world freelance job postings on Upwork, specifically from the Expensify repository. The benchmark includes approximately 1500 tasks, encompassing both individual coding contributions and managerial decision-making (proposal selection), collectively valued at $1 million based on actual payouts. A key contribution is the use of end-to-end tests, verified by experienced engineers, to assess task completion, alongside mapping performance to potential economic earnings. The authors evaluate several frontier LLMs, demonstrating that these models currently struggle with the complexity and demands of these real-world tasks.

Strengths highlighted by reviewers include the benchmark's novelty and real-world relevance, using actual freelance tasks tied to economic value (FybT, E3n6, ue7i, h9KZ). The use of end-to-end tests, verified by experienced engineers, was seen as a significant improvement over prior benchmarks relying solely on unit tests (FybT, E3n6, ue7i). The inclusion of managerial tasks (E3n6, ue7i) and the mapping of performance to monetary value (FybT, E3n6, ue7i, h9KZ) were also praised. Furthermore, the paper was generally considered clearly written (FybT) and the open-sourcing of the benchmark subset and evaluation harness was commended (E3n6).

Several potential shortcomings were raised. A primary concern, noted by FybT and E3n6, is the benchmark's reliance on a single software repository (Expensify) and platform (Upwork), potentially limiting the generalizability of the findings. Reviewers FybT and E3n6 also questioned the claim of "unbiased data collection," pointing out the inherent biases within the source platform and repository itself. The high computational cost and time required to run the benchmark were flagged by E3n6 as a barrier to broader adoption. E3n6 also noted the benchmark's focus on bug-fixing tasks (~90%) and the limitations of static end-to-end tests regarding coverage. The initial lack of confidence intervals (FybT), metrics for partial progress (FybT), evaluation beyond correctness (e.g., code quality, E3n6), and a comprehensive literature review (h9KZ) were also mentioned. Some reviewers sought further clarification on the agent framework (ue7i) and the utility of the 'User Tool' (h9KZ).

In deliberation, the authors provided substantive responses and committed to several revisions. Regarding the single-repository limitation, the authors acknowledged this constraint but justified it as a necessary step for developing a scalable process, positioning the open-sourced framework as a foundation for community expansion. They clarified that "unbiased data collection" referred to their methodology of not selectively filtering tasks based on testability, rather than implying the absence of source bias, and promised to refine this language. To address accessibility concerns due to cost, they created a smaller 'SWE-Lancer-Lite' subset. The predominance of bug-fix tasks was explained as reflective of the mature source repository chosen for its realism. While acknowledging the limitations of E2E tests and the lack of code quality evaluation, the authors argued for the current focus on correctness as a pragmatic starting point and flagged quality assessment as important future work. Crucially, the authors provided confidence intervals for key results, updated their tooling to support partial credit metrics, committed to significantly expanding the literature review and comparison with prior work, and offered clarifications on the User Tool and managerial task origins. They also planned evaluations on open-source models for the camera-ready version.

Considering the strengths and the authors' constructive engagement with reviewer feedback, the core contribution – a large-scale, economically grounded benchmark using realistic tasks and robust E2E testing – remains significant. While the limitation to a single repository is notable, the depth and realism offered represent a valuable advancement over existing benchmarks. The issues raised concerning methodology, reporting, and literature context appear to have been adequately addressed or committed to be addressed by the authors. Therefore, the substantial merits of providing a challenging, relevant, and economically interpretable benchmark for advanced software engineering AI agents seem to outweigh the remaining limitations, meeting the standards for publication at ICML.